# PTP: Boosting Stability and Performance of Prompt Tuning with Perturbation-Based Regularizer

**Lichang Chen**
University of Maryland
bobchen@cs.umd.edu

**Jiuhai Chen**
University of Maryland
jchen169@umd.edu

**Heng Huang**
University of Maryland
heng@umd.edu

**Minhao Cheng**
HKUST
minhaocheng@ust.edu.hk

## Abstract

Recent studies show that prompt tuning can better leverage the power of large language models than fine-tuning on downstream natural language understanding tasks. Nonetheless, current prompt tuning methods encounter instability during training, marked by a high variance in scores given different random seeds. In addressing this crucial issue, we uncover that the loss landscape of standard prompt tuning, when visualized, is remarkably steep, *i.e.*, minor alterations in the input data can trigger substantial fluctuations in the loss landscape, which is an essential factor that leads to the training instability. In light of this finding, we incorporate perturbation-based regularizers to temper the loss landscape within the prompt tuning process. We thus present a novel algorithm, called Prompt Tuning with Perturbation-based regularizer (PTP), that can significantly reduce training instability and concurrently enhance the performance of prompt tuning. Specifically, we design two variants of perturbation-based regularizers: one that employs random noise, and another that uses an adversarial approach. Importantly, our proposed perturbations display flexibility in both the text and embedding spaces. Extensive experiments show the effectiveness of our proposed methods in stabilizing the training. Our new algorithms improve the state-of-the-art prompt tuning methods by 1.94% and 2.34% on SuperGLUE and FewGLUE benchmarks, respectively.

## 1 Introduction

Releasing the burden of training models from scratch while keeping the outstanding performance on downstream tasks, pretrained Language Model (LM) brought NLP to a new era (Raffel et al., 2020; He et al., 2021; Shoeybi et al., 2019). Recently, inspired by the success of GPT-3 (Brown et al., 2020) on few-shot and zero-shot learning with manually created prompts, there has been a surging interest in prompting that freezes

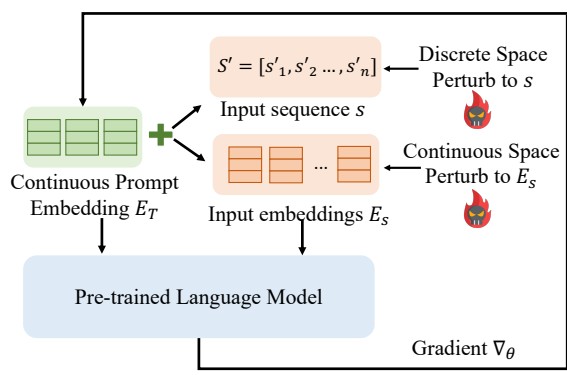

Figure 1: A simple pipeline of our **P**ropmt **T**uning with **P**erturbation-based regularizor (PTP) algorithm.

pre-trained LM and wraps up the input sequence with natural language templates. However, natural language prompts are handcrafted by experts and the performance is not comparable with fine-tuning methods. To tackle this issue, Lester et al. (2021); Li and Liang (2021) proposed prompt tuning, which prepends the input sequence with continuous embeddings and only tunes these embeddings during training. Liu et al. (2021b, 2022) verified the effectiveness of prompt tuning on natural language understanding (NLU) tasks under both few-shot learning and supervised learning settings, which is comparable to the fine-tuning methods but with much fewer ($1000\times$ less) task-specific tunable parameters. However, under different random seeds, we observe that the current prompt tuning methods suffer from a high variance of scores, which indicates they suffer from training instability issues.

To investigate the factor that causes the instability of prompt tuning, we visualize the loss landscape of the vanilla prompt tuning and observe that there exist many sharp crests, as shown in Figure 2(a), which harms the training stability. Motivated by the recent study (Chen and Hsieh, 2020) which shows that perturbation-based regularizers are powerful tools to smooth the loss landscape

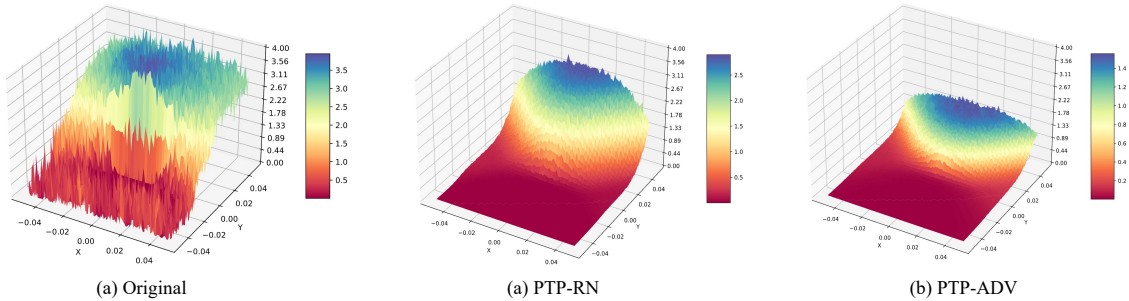

| (a) Original | (a) PTP-RN | (b) PTP-ADV |

Figure 2: The loss landscapes on different continuous prompt tuning procedures. The X-axis and Y-axis denote the magnitude of perturbations (gradient direction) and another random orthogonal direction. Z-axis represents the cross-entropy loss of the different training methods.

and stabilize the training of machine learning systems, we introduce them into prompt tuning to address the lack of stability and generalization issues. To be specific, we propose Prompt Tuning with Perturbation-based regularizer (PTP) algorithm to make the training stable and boost the performance of prompt tuning.

Specifically, we consider two kinds of perturbations in PTP, Random-Noise-based (PTP-RN) perturbation and ADVersarial-based (PTP-ADV) perturbation. PTP-RN is motivated by randomized smoothing (Cohen et al., 2019), which applies the neighborhood averaging and makes the neural network smoother while PTP-ADV is motivated by adversarial training, a method proposed to make the predictor capable of resisting the adversarial examples (Goodfellow et al., 2015) as well as boosting the clean accuracy of the predictors (Xie et al., 2020; Zhu et al., 2020). Moreover, in order to bring more flexibility and make our exploration more comprehensive, we apply perturbations to both text (discrete) and embedding (continuous) space, as depicted in Figure 1.

In the experiments, we conduct extensive experiments to evaluate our proposed algorithms, PTP-RN and PTP-ADV, on SuperGLUE (Wang et al., 2019) and FewGLUE (Schick and Schütze, 2021b) benchmark. By applying the PTP algorithm on text or embedding space to the existing prompt tuning methods, we can boost the performance of prompt tuning on SuperGLUE and FewGLUE benchmarks by 1.94% and 2.34% as well as make prompt tuning more stable. We also present a comprehensive ablation study and analysis of our algorithms with different perturbations.

Our contributions can be summarized as:

• We propose a new PTP algorithm to tackle

the training instability problem in prompt tuning, which can also boost performance. Together with the PTP algorithm, we design two types of perturbations as our implicit regularizers, which are Random-Noise-based perturbation (PTP-RN) and ADVersarial-based perturbation (PTP-ADV).

• Moreover, as depicted in Figure 1, our proposed PTP-ADV and PTP-RN can be applied to both text space and embedding space, which makes our perturbations more flexible.

• We conduct extensive experiments to evaluate the effectiveness of our algorithms on SuperGLUE and FewGLUE benchmarks. The experimental results demonstrate our proposed PTP algorithm can boost the standard performance of prompt tuning on FewGLUE and SuperGLUE by 2.34% and 1.94%, respectively. It also shows the great power of our algorithm in improving training stability.

## 2 Preliminaries and Related Work

### 2.1 Prompt Tuning

**Discrete Prompt.** Discrete prompt, also known as hard prompt (Liu et al., 2021a; Davison et al., 2019; Jiang et al., 2020; Haviv et al., 2021; Chen et al., 2023; Bao et al., 2023; Liu et al., 2023), is typically a template composed of task descriptions and original input texts. Brown et al. (2020) created templates for GPT-3 based on their introspection and make it suitable for various downstream tasks, such as machine translation, QA, *etc*. Utilizing discrete prompts, they can achieve stunning results on NLU tasks under few-shot learning settings. By employing the predefined templates and converting the tasks into cloze questions, Schick and Schütze

(2021a,b) showed that even with 'greener' backbone (Lan et al., 2020), which has 100x fewer parameters than GPT-3, they can also reach prevailing results on the few-shot version of Super-GLUE benchmark (Wang et al., 2019) (also known as FewGLUE). Liu et al. (2021b) utilized the existing discrete templates and tuned embeddings of the selected tokens, which achieves SOTA results on FewGLUE. As the few-shot scenario is common and useful, in this paper, we also test the few-shot learning ability of our PTP in our experiments.

**Continuous Prompt Tuning.** As prompts aim to boost LM's performance, it is not necessary to make tokens of prompts interpretable. Without the limit of tokens being natural words, Li and Liang (2021) proposed to prepend a series of tunable embeddings $E_T$ to the input embedding sequence as prompt and optimize them with training data, which provides in-context information for LMs to condition on. Lester et al. (2021) prepended a sequence of special tokens $T$ to the input sequence, and similarly, they tune the embeddings $E_T$ (embeddings of special tokens) on the downstream tasks. Moreover, to further leverage the power of prompt embeddings, Liu et al. (2022) presented PT2, a method that adds the trainable prompt embeddings to every layer of pretrained LMs as prefix embeddings. To keep the consistency of the notation, we also apply the same prompt embedding representation $E_T$ to represent trainable prompts in every layer of LMs.

## 2.2 Adversarial Training

**Continuous Space AT.** Over the past few years, Adversarial Training (AT) has demonstrated impressive results in improving model robustness (Goodfellow et al., 2015; Tramèr et al., 2018; Athalye et al., 2018). AT can be formulated as a min-max optimization problem:

$$\min_{\theta} \mathbb{E}_{(x_i,y_i)\sim\mathbb{D}} \left[ \max_{\|\delta\|_p \leq \epsilon} \mathcal{L}\left(\theta, x + \delta, y\right) \right], \quad (1)$$

where $\mathcal{L}$ is the loss function, $\|\cdot\|_p$ represents $\ell_p$-norm distance and $\epsilon$ denotes the perturbation budget. Madry et al. (2018) proposed PGD algorithm to compute an adversarial example (inner maximization problem) iteratively as:

$$\delta^{t+1} = \Pi_{\|\delta\|_\infty \leq \epsilon} \left( \delta^t + \alpha \operatorname{sign}\left(\nabla_\delta L(\theta, \delta^t, y)\right) \right), \quad (2)$$

where t is the iteration step; $\Pi_{\|\delta\|_\infty \leq \epsilon}$ projects perturbation $\delta$ into $\epsilon$-ball.

| | RTE | BoolQ | WiC |
|---|---|---|---|
| Acc. | $87.7 \pm 1.81$ | $83.9 \pm 0.92$ | $72.0 \pm 1.38$ |
| Var(FT) | 0.12 | 0.09 | 0.11 |
| Var(PT) | 1.45 | 0.74 | 1.16 |

Table 1: Re-implementation results of PT2 (Liu et al., 2022) with RoBERTa-large backbone on SuperGLUE benchmark. Acc.: mean accuracy. Var: variance score computed by 5 runs with different random seeds. FT: fine-tuning. PT: prompt-tuning.

Besides enhancing the robustness against adversarial examples, adversarial training has been shown great power in boosting the standard performance of image classifier (Xie et al., 2020), visual language representation learning (Gan et al., 2020), and GNN (Kong et al., 2022). In this paper, we also apply a similar idea to prompt tuning and focus on boosting its performance rather than its adversarial robustness.

**Discrete Space AT.** Different from adversarial attacks on images, in NLP attack, due to the discrete nature of text space, it is typically formulated as a combinatorial optimization problem to create adversarial input sequence $s'$, which is classically solved by heuristic search while maintaining the semantic similarity of the original input sequence $s$ (Li et al., 2020; Ren et al., 2019; Morris et al., 2020). However, the searching algorithms of adversarial attacks, such as beam search (Ebrahimi et al., 2018), greedy search based on word importance (Ren et al., 2019), and deletion-based searching (Jin et al., 2019), are usually slow because of the high computation cost on sentence encoding and large search space (Yoo et al., 2020). Yoo and Qi (2021) proposed A2T algorithm to accelerate the heuristic search process, which replaces the slow USE encoder (Li et al., 2020) with DistilBERT (Sanh et al., 2019) to calculate the cosine similarity between original input text and perturbed text, and they obtain significant speedup comparing to the textfooler (Jin et al., 2019). Thus, in this paper, we adapt the attacking algorithm in (Yoo and Qi, 2021) to generate noises in text space.

## 3 Proposed Framework

### 3.1 Preliminary

Before introducing the proposed algorithms, we first briefly describe the word to embedding process of LMs as well as the final embedding input of the continuous prompt tuning. Given the $n$ word input

**Algorithm 1:** PGD on prompt tuning

---

1 **Require:** Perturbation iteration $n$ and size $\alpha$. The bound of perturbation $\epsilon$ ;

2 **for** *epoch* = $1, \ldots, n$ **do**

3    $E_s$.requires_grad $\leftarrow$ True;

4    $\hat{y}_i \leftarrow \arg\max_y \left( \Pr\left[y | \mathcal{M}\left(I'\right)\right]\right)$;

5    $\mathcal{L}(\hat{y}, y)$.backward();

6    $E_s' \leftarrow E_s + \alpha * E_s.grad.sign()$ ;

7    $\delta = \Pi_{\|\delta\|_\infty \leq \epsilon}(E_s' - E_s)$ ;

8    $E_s' \leftarrow E_s + \delta$ ;

9    $\mathcal{M}$.zero_grad();

10 **return** $E_s'$

---

**Algorithm 2:** PTP

---

1 **Require:** Prompt embeddings $E_T$; input embeddings $E_s$; trainable parameter $\theta$ for prompt encoder $P$ and $E_T$; Training data $D$; Pre-trained LM $\mathcal{M}$; Loss function $\mathcal{L}$ ;

2 Initialize parameters $\theta$;

3 **for** *epoch* = $1, \ldots, K$ **do**

4    */* standard prompt tuning */*

5    Sample a minibatch data $(s, y)$ from $D$;

6    $\Theta$.requires_grad $\leftarrow$ True;

7    $I \leftarrow [E_T; E_s]$ ;

8    $\hat{y} \leftarrow \arg\max_y \left( \Pr\left[y | \mathcal{M}\left(I\right)\right]\right)$;

9    $\mathcal{L}(\hat{y}, y)$.backward() and update $E_T$ ;

10    */* training with perturbed data */*

11    $\Theta$.requires_grad $\leftarrow$ False;

12    Apply PTP-RN or PTP-ADV to $s$ or $E_s$;

13    $I' \leftarrow [E_T; E_s']$ ;

14    $\Theta$.requires_grad $\leftarrow$ True;

15    $\hat{y}_i \leftarrow \arg\max_y \left( \Pr\left[y | \mathcal{M}\left(I'\right)\right]\right)$;

16    $\mathcal{L}(\hat{y}, y)$.backward() and update $E_T$;

---

sequence $s = [s_1, \ldots, s_n]$ and word to embedding function $f_V$, where $V$ denotes embedding matrix, the input embedding $E_s = [e(s_1), \ldots, e(s_n)] \in \mathbb{R}^{n \times d}$ can be obtained by $E_s = f_V(s)$, where $d$ denotes the dimension of the word embedding. Let continuous prompts be $E_T = [e_t^1, \ldots, e_t^m] \in \mathbb{R}^{m \times d}$, the update of $E_T$ can be either directly, or through a reparameterization encoder $P$ like MLP/LSTM, $P(E_T) = [h_0, \ldots, h_m] \in \mathbb{R}^{m \times d}$. For simplicity, we still use $E_T$ to represent the output of $P(E_T)$, and the final input embedding sequence is written as $[E_T; E_s] \in \mathbb{R}^{(m+n) \times d}$.

The training objective of continuous prompt tuning can be formulated as:

$$\min_\theta \mathbb{E}_{(s,y) \sim \mathbb{D}} \left[ \mathcal{L}\left(\mathcal{M}\left(\theta, s, y\right)\right)\right], \quad (3)$$

where $M$ denotes LM; $\theta$ represents the trainable parameters of $E_T$ and prompt encoder $P$ while $\mathbb{D}$ is the underlying data distribution.

### 3.2 Proposed Formulation

Although continuous prompt tuning (Liu et al., 2022) could achieve comparable performance with the fine-tuning method by only using 0.1% to 3% trainable parameters, it suffers from unstable training issues: as shown in Table 1, even only changing the random seed in different runs, the final performance is very unstable. To investigate the issue, we plot the loss landscape of the vanilla prompt tuning as Figure 2 and observe sharp crests in a small local region with a small noise, which means a small perturbation on embedding space would cause a significant reduction in the final accuracy. It is also known as the training instability problem (Chen and Hsieh, 2020). To address this challenge in prompt tuning, we propose perturbation-based regularizers to force the loss landscape to be

smooth. Specifically, we introduce two versions of perturbation-based regularizers that can be formulated as follows:

$$\min_\theta \mathbb{E}_{(s,y) \sim \mathbb{D}} \left[ \mathcal{L}\left(\mathcal{M}\left(\theta, s + \delta, y\right)\right)\right], \text{ s.t.}$$
$$\text{PTP-RN: } \delta \sim \mathcal{N} \quad (4)$$
$$\text{PTP-ADV: } \delta = \max_{\|\delta\| \leq \epsilon} \mathcal{L}\left(\theta, s + \delta, y\right),$$

where $\mathcal{N}$ denotes Gaussian distribution. For PTP-RN, we minimize $\theta$ under small random perturbation, aiming to force the model to focus on perturbed pair $(s + \delta, y)$ and have a robust prediction within the neighborhood of $s$. It is related to the idea of randomized smoothing (Cohen et al., 2019), which obtains a smoother predictor via randomly averaging the neighborhood of the given function. For PTP-ADV, the perturbation $\delta$ is generated by adversarial attack algorithms such as PGD (Madry et al., 2018), A2T (Yoo and Qi, 2021), and the worst-case training loss is minimized under small perturbation bounded by $\epsilon$. The idea is motivated by adversarial training, which is usually applied as a form of adversarial defense. (Goodfellow et al., 2015; Cheng et al., 2021)

Since it is still unknown which space to inject perturbation $\delta$ is better to stabilize the training, we apply it on both text and embedding space to bring more flexibility and make our exploration more comprehensive. The perturbed sequence $s'$ or the

perturbed embedding $E'_s$ can be obtained as

$$s' = s + \delta,$$
$$E'_s = E_s + \delta, \quad (5)$$

where $s, E_s$ denote the input sequence and the input embedding, respectively. It is worth noticing that if the perturbation is on $s$ (text space), through $f_V$, the perturbed $s'$ will be converted into input embedding, which is also denoted as $E'_s$.

The main idea of our proposed formulation is that we force our algorithm to not only learn from the clean data pair $(s, y)$ but also perturbed data pair $(s + \delta, y)$ to make the training more smooth.

### 3.3 PTP-RN

#### 3.3.1 Embedding Space (RG Perturbation)

In embedding space, how to create perturbed examples is still an unsolved problem. But since the ultimate effects of PTP-RN are the only thing we care about, not the interpretability, it is feasible for us to add random-noise-based perturbation on word embeddings. Given the embeddings of the input sequence $E_s = [e(s_0), e(s_1), \dots, e(s_n)]$, where $e(s_i) \in \mathbb{R}^d$, PTP-RN in embedding space samples $\delta$ from Gaussian distribution and randomly selects some embeddings to perturb, which make sure. The perturbation $\delta$ can be formulated as:

$$E'_s = E_s + \delta, \text{s.t.}$$
$$\delta = \{\delta_1, 0, \dots, \delta_i, 0\}, \quad (6)$$

where $\delta \in \mathbb{R}^{n \times d}$ has the same length as the input embeddings; $i$ denotes the number of embeddings being perturbed and $\delta_{n=1,\dots,i} \sim \mathcal{N}(0, \sigma \mathbb{I}_d)$, with $d$ denoting the dimension of the word embedding and $\sigma$ controlling the magnitude of perturbation. We represent PTP-RN on embedding space as PTP+RG.

#### 3.3.2 Text Space (RM Perturbation)

In text space, similarly, our goal is to create label-preserving and perturbed input data to augment the training data and make the training stable. Given an input sequence $s$, PTP-RN randomly selects some tokens and converts them into [MASK]. The perturbed sequence $s'$ can be formulated as:

$$s' = \{s_0, [\text{MASK}], \dots, [\text{MASK}], s_n\}, \quad (7)$$

where we perturb $i$ tokens. It should be noticed that unlike BERT pretraining process (Vaswani et al., 2017), where the model predicts the label

of [MASK], our model will not predict anything on the tokens we mask and we just use [MASK] token as a perturbation on discrete space. PTP+RM is used to denote PTP-RN on text embedding space.

### 3.4 PTP-ADV

#### 3.4.1 Embedding Space (PGD Perturbation)

Different from previous PGD training methods which focus on improving the models' robustness, we aim to smooth the loss landscapes and boost the performance of prompt tuning by adding adversarial-based regularization. Given the embedding sequence $E_s$, PTP-ADV adopts multi-step PGD to generate perturbations on embedding space. The perturbation $\delta$ is computed iteratively as:

$$E'_s = E_s + \delta^t, \text{s.t.}$$
$$\delta^t = \Pi_{\|\delta\|_\infty \leq \epsilon} \left( \delta^{t-1} + \alpha \nabla_\delta L \right) \quad (8)$$

where $\delta^t$ denotes the $t$-th iterations of PGD perturbation and it will be added to the input embedding sequence $E_s$ after all the iterations are finished. Algorithm 1 shows the implementation details of our PGD attack on prompt tuning. PTP+PGD is applied to denote our PTP-ADV algorithm with perturbation on embedding space.

#### 3.4.2 Text Space (A2T Perturbation)

Furthermore, to enhance the flexibility of PTP-ADV and boost model generalization ability, we apply it to the text space: PTP-ADV adopts the attack algorithm in A2T (Yoo and Qi, 2021) to generate its perturbation $\delta$, which is an algorithm composed of NLP attack and adversarial training. Given the input sequence $s$, the perturbed sequence $s'$, with A2T perturbation, can be represented as:

$$s' = \{s_0, s'_1, \dots, s'_{n-1}, s_n\}, \quad (9)$$

where $s'_i$ denotes the perturbed word. For simplicity, we also call it PTP+A2T.

Algorithm 2 provides the details about the whole training process of PTP algorithm. In the standard prompt tuning, the input of LM is composed of prompt embedding $E_T$ and input embedding $E_s$, which is denoted as $I = [E_T; E_s]$. After LM gives a prediction of $I$, we backpropagate the loss to update $E_T$. In training with perturbed data part (Line 10-16, Algorithm 2), the discrete or continuous space perturbations of input data are generated by PTP-RN or PTP-ADV firstly. Then the perturbed input is employed to conduct training with original label $y$, which also plays a data-augmentation role to boost the performance of the prompt tuning.

| Method | BoolQ | | CB | | WiC | | RTE | |
|---|---|---|---|---|---|---|---|---|
| | BERT | RoBERTa | BERT | RoBERTa | BERT | RoBERTa | BERT | RoBERTa |
| FT | 77.7 | 86.9 | 94.6 | 98.2 | 74.9 | 75.6 | 70.4 | 86.6 |
| PT2 | 75.8 | 84.8 | 94.6 | 100 | 75.1 | 73.4 | 78.3 | 89.5 |
| PT2+Span-Off | 77.2 | 85.9 | 95.3 | 100 | 75.6 | 74.3 | 79.0 | 90.9 |
| PTP+A2T | 76.4 | 85.7 | 94.5 | 99.6 | 75.8 | 72.9 | 78.6 | 89.7 |
| PTP+RG | 77.3 | 86.2 | 95.8 | 99.8 | **76.7**↑1.6 | 75.5 | 79.9 | 90.6 |
| PTP+RM | 77.4 | 85.9 | 95.7 | 100 | 76.4 | 75.2 | 79.6 | 91.6 |
| PTP+PGD | **78.3**↑2.5 | **86.7**↑1.9 | **96.1**↑1.5 | **100**↑0 | 76.6 | **75.7**↑2.3 | **80.3**↑2.0 | **92.0**↑2.5 |

| Method | COPA | | MultiRC(F1a) | | ReCoRD | | WSC | |
|---|---|---|---|---|---|---|---|---|
| | BERT | RoBERTa | BERT | RoBERTa | BERT | RoBERTa | BERT | RoBERTa |
| FT | 69.0 | 94.0 | 70.5 | 85.7 | 70.6 | 89.0 | 68.3 | 63.5 |
| PT2 | 73.0 | 93.0 | 70.6 | 82.5 | 72.8 | 89.3 | 68.3 | 63.5 |
| PT2+Span-Off | 73.8 | 93.6 | 71.7 | 82.9 | 73.5 | 90.2 | 69.0 | 63.9 |
| PTP+A2T | 73.3 | 93.2 | 71.4 | 82.6 | 73.6 | 89.7 | 68.5 | 63.8 |
| PTP+RG | **75.1**↑2.1 | 93.9 | 72.6 | **84.9**↑2.4 | 74.9 | 90.5 | 69.4 | 65.0 |
| PTP+RM | 74.6 | 93.8 | 72.9 | 84.4 | 74.8 | 90.6 | 69.2 | 64.8 |
| PTP+PGD | 74.7 | **94.1**↑1.1 | **73.4**↑2.8 | 84.6 | **75.1**↑2.1 | **91.9**↑1.6 | **69.7**↑1.4 | **65.0**↑1.5 |

Table 2: Results of our proposed PTP algorithm in fully-supervised learning settings. We employ the large-size version of BERT and RoBERTa models (BERT-Large size: 335M and RoBERTa-large size: 355M, respectively). We use bold font to mark the best and red subscript to mark the improvement compared to the PT2. PT2+Span-Off: a strong data-augmentation method proposed by Shen et al. (2020).

| (Dev 32) | BoolQ | CB | WiC | RTE | MultiRC | | WSC | COPA |
|---|---|---|---|---|---|---|---|---|
| Method | (Acc.) | (F1) | (Acc.) | (Acc.) | (EM) | (F1a) | (Acc.) | (Acc.) |
| PET Best | 75.1 | 83.5 | 52.6 | 65.7 | 35.2 | 75.0 | 80.4 | 83.3 |
| PT | 77.8 | 92.3 | 56.3 | 76.5 | 36.1 | 75.0 | 84.6 | 87.0 |
| PT+Span-Off | 79.0 | 92.5 | 57.1 | 77.8 | 36.7 | 76.9 | 85.2 | 87.8 |
| PTP+RM | 79.9 | 93.2 | 58.1 | **78.6**↑2.1 | 36.2 | 78.3 | 85.9 | 88.6 |
| PTP+RG | 79.5 | **93.7**↑1.4 | 58.0 | 77.7 | 36.6 | 78.1 | 85.4 | 88.3 |
| PTP+A2T | 78.6 | 92.6 | 56.6 | 77.4 | 36.5 | 76.0 | 84.7 | 87.7 |
| PTP+PGD | **80.2**↑2.4 | 93.5 | **58.5**↑2.2 | 78.5 | **37.4**↑1.3 | **78.9**↑3.9 | **86.0**↑1.4 | **88.9**↑1.9 |
| PET(Full Dev) | 79.4 | 59.4 | 52.4 | 69.8 | 37.9 | 77.3 | 80.1 | 95.0 |
| iPET(Full Dev) | 80.6 | 92.4 | 52.2 | 74.0 | 33.0 | 74.0 | - | - |

Table 3: Results of our PTP algorithm in Few-shot learning(32 training examples) settings. PT: P-tuning (Liu et al., 2021b) and the backbone LM is alberta-xxl-v2. We use **bold** font to mark the best. The red subscript denotes the increase of our method compared with the baseline method PT. Dev 32: development set contains 32 unused examples from the training set, same as (Liu et al., 2021b). Full Dev: original development set.

## 4 Experiments

We conducted empirical studies on two popular natural language understanding (NLU) benchmarks: SuperGLUE benchmark (Wang et al., 2019) and FewGLUE benchmark (Schick and Schütze, 2021b). We tested the proposed framework in both fully-supervised and few-shot settings to verify the effectiveness of our proposed PTP-RN and PTP-ADV algorithm with perturbations on both text and embedding space. To further verify the effectiveness of our algorithm, we also conduct experiments on text summarization tasks (Narayan et al., 2018). (See Appendix §A.)

### 4.1 Experimental Setup

**NLU Dataset.** SuperGLUE benchmark (Wang et al., 2019) contains 8 challenging natural language understanding (NLU) tasks. We also include the few-shot version of SuperGLUE, FewGLUE benchmark (Schick and Schütze, 2021b) to test the ability of our algorithm, which consists of 32 training samples in each dataset on SuperGLUE. Following (Schick and Schütze, 2021b; Liu et al., 2021b), we report results on 7 of 8 NLU tasks in few-shot settings.

**Fully & Few-shot Setting.** In fully-supervised setting, the full training set of each task in SuperGLUE (Wang et al., 2019) is employed during the prompt tuning process. Besides, in the model selection part, we adopt the whole validation set. As few-shot learning ability of prompt tuning can reduce the cost of annotations in real-world applications, following (Schick and Schütze, 2021b; Liu et al., 2021b), we also test our algorithm under few-

shot settings. To be specific, we use the training set provided by FewGLUE (Schick and Schütze, 2021b), the few-shot version of SuperGLUE, containing 32 training pairs in each task. Besides, we use the same version of the development set as (Liu et al., 2021b) to select models, which are created by randomly choosing 32 unused training pairs.

**Baseline Methods.** We incorporate two popular prompt tuning methods as baselines: P-tuning (Liu et al., 2021b) (PT) and P-tuning-v2 (Liu et al., 2022) (PT2), where PT is the state-of-the-art method in FewGLUE benchmark while PT2 also achieves excellent performance in SuperGLUE benchmark. Moreover, we also include the strong data augmentation method proposed by Shen et al. (2020) and select the Span-Off, which is the best approach among the multiple augmentation approaches in their paper, for reproducing their performance on the PT and PT2. The details of hyperparameters, such as learning rate and prompt length, are deferred to Appendix B.

**Pretrained LMs.** Following the settings in (Liu et al., 2022, 2021b), we include BERT-large (Devlin et al., 2019) and RoBERTa-large (Liu et al., 2019) for fully-supervised settings and ALBERTA-xxlarge-v2 (Lan et al., 2020) for few-shot settings. To have a fair comparison with the baseline methods, for fully-supervised settings, all backbone LMs are frozen, except in fine-tuning, same as (Liu et al., 2022). For few-shot learning settings, backbone LMs are tuned with trainable prompt embeddings, same as (Liu et al., 2021b).

## 4.2 Results on Fully-supervised Setting

In fully-supervised settings, Table 2 demonstrates the results of our proposed PTP algorithm with 4 different perturbations on all 8 tasks of SuperGLUE benchmark. It is worth noticing that PTP+PGD achieves the best performance in almost all datasets except WiC (BERT), COPA(BERT), and MultiRC (RoBERTa). Overall, the best method PTP+PGD outperforms the baseline method PT2 by 1.94% (with BERT-large backbone) and 1.63% (with RoBERTa-large backbone) on average. Compared to the Span-Off (Shen et al., 2020), a strong data augmentation method, our algorithm also exhibits its advantage.

**Text vs. Embedding Perturbation.** PTP with PGD and RG perturbation on continuous space (embedding space) perform better than PTP with RM

and A2T, which indicates perturbing on continuous space is more effective than perturbing on discrete space in fully-supervised settings. As for pretrained LMs (BERT-large and RoBERTa-large), results show the superb learning ability of our PTP algorithm regardless of which backbone.

## 4.3 Results on Few-shot Setting

In few-shot learning settings, we employ FewGLUE, also known as the few-shot version of SuperGLUE. PET (Schick and Schütze, 2021b) and iPET (Schick and Schütze, 2021a) are the methods using discrete prompts. We test the previous SOTA method on FewGLUE, PT (Liu et al., 2021b) , as our baseline method and validate on the same development set (Dev 32). As illustrated in (Liu et al., 2021b), for a fair comparison, the results of PET Besst (Dev 32) are reported as removing all the additional tricks like ensemble, distillation, etc. PET (Full Dev) and iPET (Full Dev) denote the methods with the original validation set.

Our main results are shown in Table 3. PTP achieves better results than the previous state-of-the-art method PT in all 7 tasks, which verifies the effectiveness of our algorithms in few-shot NLU tasks. Especially, PTP+PGD outperforms the previous PT by 2.34% on average. Comparing the PTP+PGD (Dev 32) to the methods with the original dev set, it still has an advantage on most of the tasks (5 of 7) while the results are similar in BoolQ (better than PET with full dev set but worse than iPET). The PTP with RM and RG perturbation method also achieves remarkable improvement when compared to the baseline method PT. Moreover, PTP with A2T perturbation can also boost the performance of the baseline by a small margin.

## 4.4 Results on Improving Training Stability

In addition to improved generalization performance, the proposed method could stabilize the training process. Figure 2 provides strong evidence that our proposed PTP-RN and PTP-ADV training methods have a much smoother loss landscape compared to the vanilla prompt tuning. For the few-shot learning setting, we demonstrate the results in Table 7. It could be seen that all our PTP methods have smaller variances than the baseline method. Specifically, PTP+PGD has the smallest variance in 5 runs, which indicates its training and generalization stability. Compared with the PTP-RN methods (RG, RM), PTP-ADV methods (A2T,

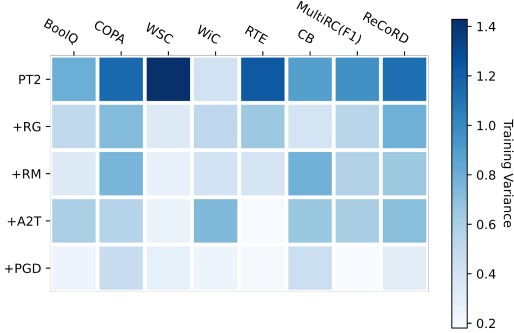

Figure 3: The variance of the scores on the dev sets from SuperGLUE Benchmark. We compute it on 5 runs with different random seeds to report. The reported experiments are all using BERT-large models as backbones.

PGD) achieve smaller variance. We also conduct the experiments under fully-supervised learning settings in Figure 3. It shows that in all 8 tasks from SuperGLUE benchmark, our proposed method can still reduce the training variance of different runs with the same hyperparameter except the seeds.

## 5 Ablation Study

In this section, we conduct comprehensive ablation studies of our perturbation methods under both fully-supervised and few-shot learning settings.

**RG Perturbation.** We investigate the strength of Gaussian noise $\delta$ and number of perturbed embeddings $i$ affects the performance (see Eq. (6)). Specifically, we run experiments with $\sigma$ , which is the variance of the added Gaussian noise, from $\{10^{-4}, 10^{-3}, 10^{-2}\}$ and the number of word embeddings perturbed, which is denoted as $i$ in Eq. (6), from $\{1, 5, 10, 20\}$. Under the fully-supervised learning setting, we report the results on COPA, RTE, and WiC tasks in Figure 4. The results show the appropriate $\sigma$ is supposed to be $10^{-2}$ and the number of perturbed embeddings to be 5. With large $\sigma$ and large $i$, PTP+RG is more likely to fail in comparison to the baseline method. Under few-shot learning settings, we select results in MultiRC task to report, as shown in Table 8. The encouraging result also demonstrates that the best choice of $\sigma$ and number of embeddings perturbed is $10^{-3}$ and 5, respectively.

**PGD Perturbation.** We investigate how different $\alpha$ (See Eq. (8)) and PGD iterations affect the performance. We run experiments with $\alpha$ from $\{10^{-4}, 10^{-3}, 10^{-2}\}$ and PGD iterations from 1 to

5. Under fully-supervised learning settings, we present the the results of COPA dataset in Table 9. It shows that large $\alpha$ in PGD will be detrimental to the performance. Under few-shot learning settings, Figure 5 demonstrates the results of different $\alpha$ and different iterations of PTP+PGD on few-shot settings. In all 3 datasets, when $\alpha$ is $10^{-3}$, not too small nor too large, and PGD iters is 4, PTP+PGD can achieve outstanding performance.

**RM Perturbation.** We investigate how different numbers of random [MASK] inserted affect the PTP. Formally, the number of [MASK] inserted is defined as $i$ in Eq. (7). We conduct experiments with $i$ from 1 to 10. Under few-shot learning settings, Figure 6 presents the results of PTP with RM perturbation on FewGLUE (MultiRC and RTE dataset). We observe that RM perturbation can boost the performance substantially in few-shot settings and the best choice of $i$ is 8. Under fully-supervised settings, Table 10 presents the ablation of RM perturbation. It also shows that a large number of [MASK] inserted harms the performance, especially when the backbone is RoBERTa.

**A2T Perturbation.** We investigate how the minimum cosine similarity between normal input $s$ and perturbed input $s'$ of A2T perturbation affects the results. We run experiments with minimum cosine similarity from $\{0.2, 0.4, 0.6, 0.8\}$ and show results in Table 11. It indicates small similarity may cause damage to the standard performance because the perturbation can be too large in this case.

## 6 Conclusion

In this paper, we first investigated the training instability issues on prompt tuning, which has a precipitous loss landscape in its visualization. To tackle the problem, we proposed PTP-RN and PTP-ADV algorithms, which include four different perturbations (RG, RM, ADV, A2T) on both discrete and continuous spaces, to smooth the loss landscape and make the training stable. Furthermore, our algorithms are also capable of boosting the performance of prompt tuning. The extensive experiments validate the effectiveness of our proposed algorithms on NLU tasks under both fully-supervised and few-shot settings.

## 7 Limitation

In this paper, we introduce the PTP algorithm, which enhances the stability and performance of

prompt tuning. It should be noted, however, that our approach is exclusively relevant to continuous prompt tuning, and cannot be used to improve other models like GPT-3 or ChatGPT, which are only accessible via their APIs. Another limitation of our algorithm is that we do not provide a theoretical analysis.

# 8 Acknowledgement

LC Chen and H Huang were partially supported by NSF IIS 1838627, 1837956, 1956002, 2211492, CNS 2213701, CCF 2217003, DBI 2225775.

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

# APPENDIX

## A    PTP on Seq2seq tasks

In this section, we demonstrate the versatility of our method by applying it to text summarization. We utilize a different language model backbone, BART-large (Lewis et al., 2019), which is an encoder-decoder model. Our experiment setting follows Li and Liang (2021), including the hyperparameter details and dataset split. We also use their results as our baseline.

| Method | R-1 | R-2 | R-L |
|---|---|---|---|
| Prefix-tuning | 43.80 | 20.93 | 37.25 |
| PTP + A2T | 44.19 | 21.06 | 37.28 |
| PTP + RG | 44.21 | 21.15 | 37.84 |
| PTP + RM | 44.12 | 21.31 | 37.91 |
| PTP + PGD | **44.35** | **21.42** | **38.23** |

Table 4: Text summarization on XSUM. Prefix-tuning is the baseline method.

As for the number of parameters, we use the same as the prefix tuning, with only 2% of the fine-tuning required. As Tabel 4 shows, Our algorithm can also achieve better performance on all three scores, ROUGE-1, ROUGE-2, and ROUGE-L.

## B    Details of learning rate and prompt length

| Tasks | LR1 | LR2 | PL1 | PL2 |
|---|---|---|---|---|
| BoolQ | 1e-3 | 5e-3 | 40 | 16 |
| COPA | 5e-3 | 7e-3 | 16 | 16 |
| RTE | 1e-2 | 5e-3 | 20 | 128 |
| WSC | 3e-3 | 7e-3 | 16 | 8 |
| CB | 7e-3 | 9e-3 | 16 | 16 |
| MultiRC | 1e-4 | 3e-3 | 40 | 20 |
| ReCoRD | 3e-4 | 5e-3 | 16 | 40 |
| WiC | 1e-4 | 5e-3 | 20 | 16 |

Table 5: Prompt Length and Learning Rate details for 8 tasks on SuperGLUE. LR1, PL1: learning rate and prompt length for continuous prompts with BERT-large backbone. LR2 and PL2: learning rate and prompt length for prompts with RoBERTa-large backbone.

Under fully-supervised settings, the prompt length and learning rate details are presented as Table 5. Under few-shot learning settings, we report it as Table 6. The prompt length is exactly

| Tasks | Learning Rate |
|---|---|
| BoolQ | 5e-5 |
| RTE | 5e-5 |
| WiC | 1e-5 |
| WSC | 5e-5 |
| COPA | 1e-5 |
| MultiRC | 1e-4 |
| CB | 1e-5 |

Table 6: Prompts' Learning Rate details for 7 tasks in FewGLUE.

the same as the PT (Liu et al., 2021b), thus we ignore it here. The hyperparameters of Span-Off are followed as the original paper (Shen et al., 2020). To ensure a fair comparison, we make the training epochs of Span-Off the same as our algorithm.

## C    Supplement for Experiment

This section includes the tables and figures as a supplement to our experiment and ablations. Table 7 demonstrates the comparison of the variance of different training methods on FewGLUE benchmark. Table 8 presents the ablation of our algorithm with RG perturbation on MultiRC (Khashabi et al., 2018) task. We show the ablation of PGD perturbation in Table 9. The ablation of RM perturbation is presented as Table 10. Table 11 shows the ablation of our proposed PTP+A2T training algorithm.

Figure 4 shows the ablation of RG perturbations on WiC (Pilehvar and Camacho-Collados, 2018), RTE (Dagan et al., 2005), and COPA (Gordon et al., 2012) datasets under fully-supervised learning setting while Figure 5 presents the ablation of our PTP+PGD training method on WiC (Pilehvar and Camacho-Collados, 2018), BoolQ (Clark et al., 2019) and WSC (Levesque et al., 2012) datasets under few-shot learning settings. We show the ablation of our proposed PTP+RM algorithm as Figure 6 on FewGLUE benchmark.

All experiments are conducted on servers with RTX A6000 GPUs, each having 48GB of memory.

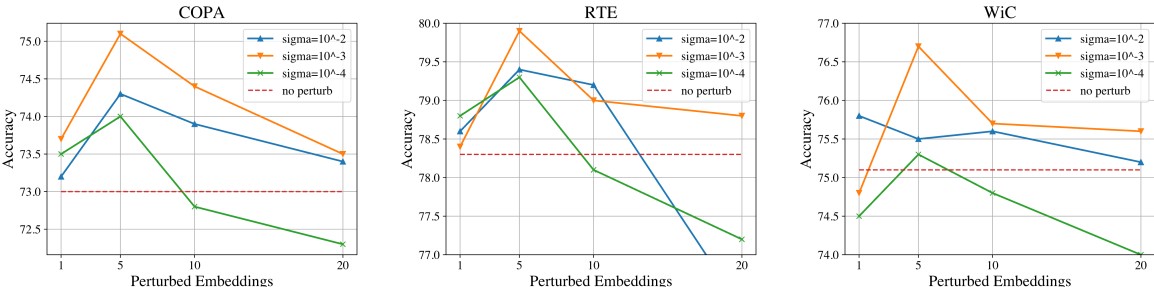

Figure 4: Performance of PTP+RG on SuperGLUE (WiC, RTE, COPA datasets) with $\sigma$ from $\{10^{-4}, 10^{-3}, 10^{-2}\}$ and perturbed embeddings from $\{1, 5, 10, 20\}$. The dashed red line represents the performance of baseline method PT2 with BERT-large as backbone LM.

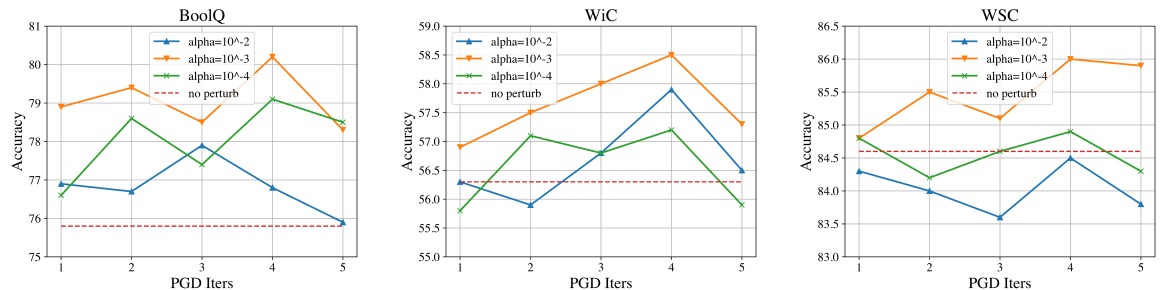

Figure 5: Performance of PTP+PGD on FewGLUE (WiC, BoolQ, WSC datasets) with different $\alpha$ and PGD iterations. The dashed red line represents the performance of the baseline method PT (Liu et al., 2021b). It shows the best $\alpha$ and PGD iterations are $10^{-3}$ and 4, respectively.

| Tasks | RTE | WSC | WiC | BoolQ |
|-------|-----|-----|-----|-------|
| PT | 1.89 | 1.68 | 1.77 | 1.45 |
| +RG | 0.81 | 0.78 | 0.91 | 0.56 |
| +A2T | 0.45 | 0.43 | $\mathbf{0.39}_{\downarrow 1.38}$ | 0.59 |
| +RM | 0.68 | 0.95 | 0.87 | 0.65 |
| +PGD | $\mathbf{0.35}_{\downarrow 1.54}$ | $\mathbf{0.31}_{\downarrow 1.37}$ | 0.47 | $\mathbf{0.43}_{\downarrow 1.02}$ |

Table 7: The variance of the scores on the dev sets of RTE, COPA and BoolQ from the FewGLUE benchmark. We compute it on 5 runs with different random seeds (other hyper-parameter are the same). We employ bold font to denote the smallest deviation in each task and blue font to denote the decrease when compared to PT.

| MultiRC | E=1 | E=5 | E=10 | E=20 |
|---------|-----|-----|------|------|
| $\sigma$=1e-2 | 75.2 | $74.3_{\downarrow 0.7}$ | 75.8 | 74.7 |
| $\sigma$=1e-3 | 77.3 | $\mathbf{78.1}_{\uparrow 3.1}$ | 77.1 | 76.6 |
| $\sigma$=1e-4 | 76.9 | 77.2 | 76.8 | 75.5 |

Table 8: Results of different $\sigma$ in RG perturbation (MultiRC task). The baseline method is PT2 and its performance is 75.0. E denotes number of embeddings that are perturbed. We mark the best and the worst.

| COPA | t=1 | 2 | 3 | 4 | 5 |
|------|-----|---|---|---|---|
| $\alpha$=1e-2 | 70.3 | 72.1 | 72.0 | 71.0 | $69.0_{\downarrow 4.0}$ |
| 1e-3 | 73.4 | 70.8 | 72.9 | 73.5 | 73.2 |
| 1e-4 | 73.1 | 73.4 | $\mathbf{74.7}_{\uparrow 1.7}$ | 73.8 | 72.5 |

Table 9: Results of PTP+PGD on fully-supervised COPA dataset. The baseline method is PT2 (Liu et al., 2022), whose accuracy is 73.0 . The backbone LM employed is BERT-large. $\alpha$ is the perturbation size while $t$ is PGD iterations (see Eq. (8)).

| Dataset | LM | 1 | 2 | 3 | 4 | 5 | 6 | 7 | 8 | 9 | 10 |
|---|---|---|---|---|---|---|---|---|---|---|---|
| BoolQ | BERT | -0.12 | +0.50 | **+1.57** | +0.65 | +1.35 | +1.14 | +1.12 | +0.24 | +0.25 | -0.38 |
| (Full) | RoBERTa | +0.31 | +0.76 | **+1.10** | +0.88 | +0.69 | -0.36 | +0.19 | -0.43 | -0.67 | -0.36 |

Table 10: The results of different $i$ in Eq. (7) (number of [MASK] randomly inserted into input sequence as perturbation) . We select BoolQ with fully-supervised settings to report. The reported increase or decrease is compared to the baseline method PT2.

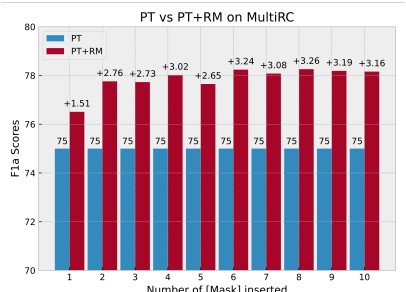 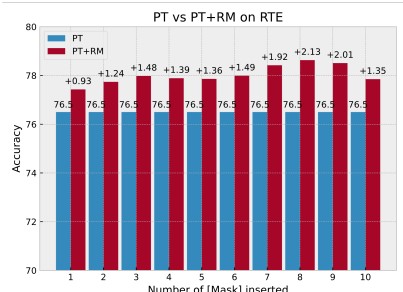

Figure 6: The results of different $i$ in Eq. (7) (number of [MASK] randomly inserted into input sequence as perturbation). We select MultiRC and RTE tasks with Few-shot setting to report.

| Cosine Sim | 0.2 | 0.4 | 0.6 | 0.8 |
|---|---|---|---|---|
| BoolQ(Full) | -0.83 | -0.46 | +0.61 | +0.52 |
| BoolQ(Few) | -0.96 | -0.13 | +0.78 | +0.84 |
| MultiRC(Full) | -0.77 | -0.35 | +0.25 | +0.71 |
| MultiRC(Few) | -0.59 | -0.54 | +1.03 | +0.68 |

Table 11: The results of different minimum cosine similarity in A2T perturbation. Full: fully-supervised learning setting. Few: few-shot learning setting.