# OpenReview forum: "PTP: Boosting Stability and Performance of Prompt Tuning with Perturbation-Based Regularizer"
_EMNLP/2023/Conference — EMNLP 2023 Main_

### Official Review · Reviewer_uMXZ · 2023-08-03

**Soundness:** 3

**Excitement:**

3: Ambivalent: It has merits (e.g., it reports state-of-the-art results, the idea is nice), but there are key weaknesses (e.g., it describes incremental work), and it can significantly benefit from another round of revision. However, I won't object to accepting it if my co-reviewers champion it.

**Paper Topic And Main Contributions:**

This paper proposes “Prompt Tuning with Perturbation-based regularizer” (PTP), that can significantly reduce training instability and concurrently enhances the performance of prompt tuning. Two types of perturbations are designed as implicit regularizers, random-noise-based perturbation (PTP-RN) and adversarial-based perturbation (PTP-ADV). These two regularizers can be applied to both text space and embedding space. Experiments under super-glue and few-glue benchmarks show that PTP can significantly improve the performance of standard prompt tuning.

**Questions For The Authors:**


Detailed comments and questions:
1.	Any detailed comparison of prefix-tuning with PTP? Where prefix-tuning appends past-key-value information by virtual tokens to the middle layers, different from current embedding space and text space. Or, any idea of combination them?
2.	Can you show several examples of perturbed datasets? Just wonder what are exactly perturbed when used in lines 10 to 16 in Algorithm 2.


**Reasons To Accept:**

Strong:
1.	PTP idea with two types of implicit regularizers are impressive;
2.	Significantly better results are achieved.


**Reasons To Reject:**

Weak:
1.	The approach is mainly constrained by the quality of “perturbed data” and further detained methods of construction these datasets together with the consequent performance change are required, in multilingual and multi-task scenarios.
2.	Not quite clear about why PTP are not applied to the middle transformer layers in LLMs, but only at embedding space and text space?


**Reproducibility:**

3: Could reproduce the results with some difficulty. The settings of parameters are underspecified or subjectively determined; the training/evaluation data are not widely available.

**Reviewer Confidence:**

3: Pretty sure, but there's a chance I missed something. Although I have a good feel for this area in general, I did not carefully check the paper's details, e.g., the math, experimental design, or novelty.

---

### Official Review · Reviewer_uB8g · 2023-08-03

**Soundness:** 3

**Excitement:**

3: Ambivalent: It has merits (e.g., it reports state-of-the-art results, the idea is nice), but there are key weaknesses (e.g., it describes incremental work), and it can significantly benefit from another round of revision. However, I won't object to accepting it if my co-reviewers champion it.

**Missing References:**

* [Chen et al., 2022](https://aclanthology.org/2022.emnlp-main.168): Revisiting Parameter-Efficient Tuning: Are We Really There Yet?
This paper investigates the instability of prompt tuning and offers potential causes and also remedies to mitigate it.
Certain findings of that paper contradict the ones in this paper (e.g. Chen et al. found that prompt-tuning lags far behind fine-tuning), which needs to be discussed in the paper.

**Paper Topic And Main Contributions:**

The paper deals with improving the stability of prompt tuning.
The idea is to apply data augmentation (both in the continuous and the discrete space) via either adding random perturbations to the input or employing adversarial training for the real data points and optimize towards the proper prompts over the augmented set of data points.

The paper gives the impression that the instability of prompt tuning was not investigated earlier (e.g. line 281 says 'Since we are the first to investigate the training stability issue of the prompt tuning'), however, this statement is not true as [Chen et al. (2022)](https://aclanthology.org/2022.emnlp-main.168) have already tackled the problem and also identified potential causes (and remedies).

The data augmentation comes with reasonable computational burden (especially the one incorporating adversarial training).
If the less resource intensive prompt tuning (without augmentation) or traditional fine-tuning was performed longer and with the same computational budget as the proposed approach enjoying the additional benefit of using augmented data points, their results might as well improve and become more stable.
It is not clear if the different methods were ensured to have (roughly) the same computational budget.

Even though the main motivation of the paper is to reduce the variance of prompt-tuning methods, that aspect is not well displayed in the paper among the experimental results.
As for the few shot evaluation, the variance scores are reported only in the appendix, and the variance scores are only reported for 4 of the 7 investigated tasks.
These results are either incomplete at best, or cherry-picked at worst.
For the fully-supervised setting, variances for all the tasks are reported, but only in the form of a heatmap, which is [hard to interpret and can be misleading](https://link.springer.com/chapter/10.1007/978-3-642-02574-7_4) without the actual figures being reported.

**Questions For The Authors:**

A) Did the different approaches used (roughly) the same amount of compute?   If not, how much would the results differ if the different approaches were provided with the same amount of computation?

**Reasons To Accept:**

Prompt tuning is widely used and notoriously unstable, hence the paper addresses a timely problem.

**Reasons To Reject:**

The paper misframes its contributions as it is not the first to identify and try to mitigate the instability of prompt tuning.
As data augmentation and prompt tuning with the augmented data arguably comes with additional computational burden, it is not not clear if simply training longer without data augmentation can achieve similar results.
This question is especially important as the suggestion from prior work for making prompt tuning more stable is to reduce the number of tunable parameters and to train longer.

**Reproducibility:**

4: Could mostly reproduce the results, but there may be some variation because of sample variance or minor variations in their interpretation of the protocol or method.

**Reviewer Confidence:**

4: Quite sure. I tried to check the important points carefully. It's unlikely, though conceivable, that I missed something that should affect my ratings.

**Typos Grammar Style And Presentation Improvements:**

Adding the actual variance figures to the heatmap in Figure 3 (or simply using a table) would help assessing the variability of the different approaches.

### Typos
line 434: ALBERTA -> ALBERT
line 458: are perform better -> perform better
line 472: mind thte extra whitespace before the coma in ' , as '
line 475: PET Besst -> PET Best
line 497: stable -> stabilize

---

### Official Review · Reviewer_RZJH · 2023-08-05

**Typos Grammar Style And Presentation Improvements:** Lines 309-310
**Soundness:** 4

**Excitement:**

4: Strong: This paper deepens the understanding of some phenomenon or lowers the barriers to an existing research direction.

**Missing References:**

Papers on neural model/vanilla fine-tuning instability, e.g.,

* Fine-Tuning Pretrained Language Models: Weight Initializations, Data Orders, and Early Stopping, Dodge et al., 2020
* On the Reproducibility of Neural Network Predictions, Bhojanapalli et al., 2021
* Similarity and Matching of Neural Network Representations, Csiszárik et al., 2021
* Noise Stability Regularization for Improving BERT Fine-tuning, Hua et al., 2021
* On the Stability of Fine-tuning BERT: Misconceptions, Explanations, and Strong Baselines, Mosbach et al., 2021
* Revisiting Few-sample BERT Fine-tuning, Zhang et al., 2021
* Measuring the Instability of Fine-Tuning, Du and Nguyen, 2023

**Paper Topic And Main Contributions:**

This paper studied the problem of the instability of prompt-tuning, i.e., different fine-tuning runs with only random seed difference result in very different performance. The author observed that there were many sharp crest in the loss landscape of prompt-tuning, and suspected this phenomenon to be the origin. To mitigate this instability, this paper proposed to add perturbation-based regularizers for prompt-tuning (PTP). Concretely, two different types of perturbations on both input and embedding spaces were considered: random noises and adversarial-based perturbation. Experiments on FewGLUE and SuperGLUE showed that PTP could both stabilize fine-tuning and enhance fine-tuned model’s performance.

**Reasons To Accept:**

* The writing is super clear. I can easily follow the contents, even for relatively complex concepts. Can see that the authors put efforts on revising.
* The proposed algorithm is simple yet effective, and allows for flexibility for future studies (e.g., where to inject noises, which kinds of adversarial noises to inject, etc.).
* I like the reasoning from sharp crests to regularization — and the performance is convincing.

**Reasons To Reject:**

* This paper didn’t discuss the research line of the stability of vanilla fine-tuning or generally neural models, which I believe is important and relevant (relevant papers are mentioned in missing references).
* The authors motivate the work with stability; however, stability is a rather small theme in experiments (Table 7 is in Appendix C). I would sugest the authors to better test the motivating theory (e.g., how is the loss landscape under PTP? Do we observe better stability after maunally injecting noise during inference?), and measure instability (e.g. Churn, Bhijanapalli et al., 2021; Prediction and representation instability, Du and Nguyen, 2023; both papers see missing references). I am happy to raise my soundness to 4 or 5 if the authors can put more effort here.

**Reproducibility:**

5: Could easily reproduce the results.

**Reviewer Confidence:**

5: Positive that my evaluation is correct. I read the paper very carefully and I am very familiar with related work.

---

### Meta-Review · Area_Chair_w7kw · 2023-09-14

**Recommendation:** 4

**Metareview:**

This paper address the problem of instability of prompt tuning approaches. The paper addresses an important problem as prompt-tuning is a very popular choice to fine-tune LLMs for specific tasks without incurring in prohibitive costs. The author propose an approach to introduce perturbations in the input or embedding spaces in order to stabilize the model. The reviewers agree that the idea of the paper is sound and that the contribution is relevant to the problem. The experiments proposed by the authors are comprehensive and demonstrate that the approach is working to reduce the instability of prompt tuning training.

---

### Decision · Program_Chairs · 2023-10-07

**Decision:**

Accept-Main

**Comment:**

This paper address the problem of instability of prompt tuning approaches. The paper addresses an important problem as prompt-tuning is a very popular choice to fine-tune LLMs for specific tasks without incurring in prohibitive costs. The author propose an approach to introduce perturbations in the input or embedding spaces in order to stabilize the model. The reviewers agree that the idea of the paper is sound and that the contribution is relevant to the problem. The experiments proposed by the authors are comprehensive and demonstrate that the approach is working to reduce the instability of prompt tuning training.